# Aggregating Optimistic Planning Trees for Solving Markov Decision Processes

**Gunnar Kedenburg**
INRIA Lille - Nord Europe / idalab GmbH
gunnar.kedenburg@inria.fr

**Raphaël Fonteneau**
University of Liège / INRIA Lille - Nord Europe
raphael.fonteneau@ulg.ac.be

**Rémi Munos**
INRIA Lille - Nord Europe / Microsoft Research New England
remi.munos@inria.fr

## Abstract

This paper addresses the problem of online planning in Markov decision processes using a randomized simulator, under a budget constraint. We propose a new algorithm which is based on the construction of a forest of planning trees, where each tree corresponds to a random realization of the stochastic environment. The trees are constructed using a "safe" optimistic planning strategy combining the optimistic principle (in order to explore the most promising part of the search space first) with a safety principle (which guarantees a certain amount of uniform exploration). In the decision-making step of the algorithm, the individual trees are aggregated and an immediate action is recommended. We provide a finite-sample analysis and discuss the trade-off between the principles of optimism and safety. We also report numerical results on a benchmark problem. Our algorithm performs as well as state-of-the-art optimistic planning algorithms, and better than a related algorithm which additionally assumes the knowledge of all transition distributions.

## 1 Introduction

Adaptive decision making algorithms have been used increasingly in the past years, and have attracted researchers from many application areas, like artificial intelligence [16], financial engineering [10], medicine [14] and robotics [15]. These algorithms realize an adaptive control strategy through interaction with their environment, so as to maximize an a priori performance criterion.

A new generation of algorithms based on look-ahead tree search techniques have brought a breakthrough in practical performance on planning problems with large state spaces. Techniques based on planning trees such as Monte Carlo tree search [4, 13], and in particular the UCT algorithm (UCB applied to Trees, see [12]) have allowed to tackle large scale problems such as the game of Go [7]. These methods exploit that in order to decide on an action at a given state, it is not necessary to build an estimate of the value function everywhere. Instead, they search locally in the space of policies, around the current state.

We propose a new algorithm for planning in Markov Decision Problems (MDPs). We assume that a limited budget of calls to a randomized simulator for the MDP (the generative model in [11]) is available for exploring the consequences of actions before making a decision. The intuition behind our algorithm is to achieve a high exploration depth in the look-ahead trees by planning in fixed realizations of the MDP, and to achieve the necessary exploration width by aggregating a forest of planning trees (forming an approximation of the MDP from many realizations). Each of the trees is developed around the state for which a decision has to be made, according to the principle of optimism in the face of uncertainty [13] combined with a safety principle.

We provide a finite-sample analysis depending on the budget, split into the number of trees and the number of node expansions in each tree. We show that our algorithm is consistent and that it identifies the optimal action when given a sufficiently large budget. We also give numerical results which demonstrate good performance on a benchmark problem. In particular, we show that our algorithm achieves much better performance on this problem than OP-MDP [2] when both algorithms generate the same number of successor states, despite the fact that OP-MDP assumes knowledge of all successor state probabilities in the MDP, whereas our algorithm only samples states from a simulator.

The paper is organized as follows: first, we discuss some related work in section 2. In section 3, the problem addressed in this paper is formalized, before we describe our algorithm in section 4. Its finite-sample analysis is given in section 5. We provide numerical results on the inverted pendulum benchmark in section 6. In section 7, we discuss and conclude this work.

## 2   Related work

The optimism in the face of uncertainty paradigm has already lead to several successful results for solving decision making problems. Specifically, it has been applied in the following contexts: multi-armed bandit problems [1] (which can be seen as single state MDPs), planning algorithms for deterministic systems and stochastic systems [8, 9, 17], and global optimization of stochastic functions that are only accessible through sampling. See [13] for a detailed review of the optimistic principle applied to planning and optimization.

The algorithm presented in this paper is particularly closely related to two recently developed online planning algorithms for solving MDPs, namely the OPD algorithm [9] for MDPs with deterministic transitions, and the OP-MDP algorithm [2] which addresses stochastic MDPs where all transition probabilities are known. A Bayesian adaptation of OP-MDP has also been proposed [6] for planning in the context where the MDP is unknown.

Our contribution is also related to [5], where random ensembles of state-action independent disturbance scenarios are built, the planning problem is solved for each scenario, and a decision is made based on majority voting. Finally, since our algorithm proceeds by sequentially applying the first decision of a longer plan over a receding horizon, it can also be seen as a Model Predictive Control [3] technique.

## 3   Formalization

Let $(\mathcal{S}, \mathcal{A}, p, r, \gamma)$ be a Markov decision process (MDP), where the set $\mathcal{S}$ and $\mathcal{A}$ respectively denote the state space and the finite action space, with $|\mathcal{A}| > 1$, of the MDP. When an action $a \in \mathcal{A}$ is selected in state $s \in \mathcal{S}$ of the MDP, it transitions to a successor state $s' \in \mathcal{S}(s, a)$ with probability $p(s'|s, a)$. We further assume that every successor state set $\mathcal{S}(s, a)$ is finite and their cardinality is bounded by $K \in \mathbb{N}$. Associated with the transition is an deterministic instantaneous reward $r(s, a, s') \in [0, 1]$.

While the transition probabilities may be unknown, it is assumed that a randomized simulator is available, which, given a state-action pair $(s, a)$, outputs a successor state $s' \sim p(\cdot|s, a)$. The ability to sample is a weaker assumption than the knowledge of all transition probabilities. In this paper we consider the problem of planning under a budget constraint: only a limited number of samples may be drawn using the simulator. Afterwards, a single decision has to be made.

Let $\pi : \mathcal{S} \to \mathcal{A}$ denote a deterministic policy. Define the value function of the policy $\pi$ in a state $s$ as the discounted sum of expected rewards:

$$v^\pi : \mathcal{S} \to \mathbb{R}, \ \ v^\pi : s \mapsto \mathbb{E}\left[\sum_{t=0}^\infty \gamma^t r(s_t, \pi(s_t), s_{t+1})\big|s_0 = s\right], \tag{1}$$

where the constant $\gamma \in (0, 1)$ is called the discount factor. Let $\pi^*$ be an optimal policy (i.e. a policy that maximizes $v^\pi$ in all states). It is well known that the optimal value function $v^* := v^{\pi^*}$ is the

solution to the Bellman equation

$$\forall s \in \mathcal{S}: \ v^*(s) = \max_{a \in \mathcal{A}} \sum_{s' \in \mathcal{S}(s,a)} p(s'|s,a)\left(r(s,a,s') + \gamma v^*(s')\right).$$

Given the action-value function $Q^* : (s,a) \mapsto \sum_{s' \in \mathcal{S}(s,a)} p(s'|s,a)(r(s,a,s') + \gamma v^*(s'))$, an optimal policy can be derived as $\pi^* : s \mapsto \operatorname{argmax}_{a \in \mathcal{A}} Q^*(s,a)$.

## 4  Algorithm

We name our algorithm ASOP (for "Aggregated Safe Optimistic Planning"). The main idea behind it is to use a simulator to obtain a series of deterministic "realizations" of the stochastic MDP, to plan in each of them individually, and to then aggregate all the information gathered in the deterministic MDPs into an empirical approximation to the original MDP, on the basis of which a decision is made.

We refer to the planning trees used here as single successor state trees (S3-trees), in order to distinguish them from other planning trees used for the same problem (e.g. the OP-MDP tree, where all possible successor states are considered). Every node of a S3-tree represents a state $s \in \mathcal{S}$, and has at most one child node per state-action $a$, representing a successor state $s' \in \mathcal{S}$. The successor state is drawn using the simulator during the construction of the S3-tree.

The planning tree construction, using the SOP algorithm (for "Safe Optimistic Planning"), is described in section 4.1. The ASOP algorithm, which integrates building the forest and deciding on an action by aggregating the information in the forest, is described in section 4.2.

### 4.1  Safe optimistic planning in S3-trees: the SOP algorithm

SOP is an algorithm for sequentially constructing a S3-tree. It can be seen as a variant of the OPD algorithm [9] for planning in deterministic MDPs. SOP expands up to two leaves of the planning tree per iteration. The first leaf (the *optimistic* one) is a maximizer of an upper bound (called b-value) on the value function of the (deterministic) realization of the MDP explored in the S3-tree. The b-value of a node $x$ is defined as

$$b(x) := \sum_{i=0}^{d(x)-1} \gamma^i r_i + \frac{\gamma^{d(x)}}{1-\gamma} \tag{2}$$

where $(r_i)$ is the sequence of rewards obtained along the path to $x$, and $d(x)$ is the depth of the node (the length of the path from the root to $x$). Only expanding the optimistic leaf would not be enough to make ASOP consistent; this is shown in the appendix. Therefore, a second leaf (the *safe* one), defined as the shallowest leaf in the current tree, is also expanded in each iteration. A pseudo-code is given as algorithm 1.

---

**Algorithm 1:** SOP

---

**Data**: The initial state $s_0 \in \mathcal{S}$ and a budget $n \in \mathbb{N}$
**Result**: A planning tree $T$
Let $T$ denote a tree consisting only of a leaf, representing $s_0$.
Initialize the cost counter $c := 0$.
**while** $c < n$ **do**
    Form a subset of leaves of $T$, $L$, containing a leaf of minimal depth, and a leaf of maximal b-value
    (computed according to (2); the two leaves can be identical).
    **foreach** $l \in L$ **do**
        Let $s$ denote the state represented by $l$.
        **foreach** $a \in \mathcal{A}$ **do**
            **if** $c < n$ **then**
                Use the simulator to draw a successor state $s' \sim p(\cdot|s,a)$.
                Create an edge in $T$ from $l$ to a new leaf representing $s'$.
                Let $c := c + 1$.
**return** $T$

---

## 4.2 Aggregation of S3-trees: the ASOP algorithm

ASOP consists of three steps. In the first step, it runs independent instances of SOP to collect information about the MDP, in the form of a forest of S3-trees. It then computes action-values $\hat{Q}^*(s_0, a)$ of a single "empirical" MDP based on the collected information, in which states are represented by forests: on a transition, the forest is partitioned into groups by successor states, and the corresponding frequencies are taken as the transition probabilities. Leaves are interpreted as absorbing states with zero reward on every action, yielding a trivial lower bound. A pseudo-code for this computation is given as algorithm 2. ASOP then outputs the action

$$\hat{\pi}(s_0) \in \operatorname*{argmax}_{a \in \mathcal{A}} \hat{Q}^*(s_0, a).$$

The optimal policy of the empirical MDP has the property that the empirical lower bound of its value, computed from the information collected by planning in the individual realizations, is maximal over the set of all policies. We give a pseudo-code for the ASOP algorithm as algorithm 3.

---

**Algorithm 2:** ActionValue

---

**Data**: A forest $F$ and an action $a$, with each tree in $F$ representing the same state $s$
**Result**: An empirical lower bound for the value of $a$ in $s$
Let $E$ denote the edges representing action $a$ at any of the root nodes of $F$.
**if** $E = \emptyset$ **then**
 | **return** $0$
**else**
 | Let $F$ be the set of trees pointed to by the edges in $E$.
 | Enumerate the states represented by any tree in $F$ by $\{s'_i : i \in I\}$ for some finite $I$.
 | **foreach** $i \in I$ **do**
 |  | Denote the set of trees in $F$ which represent $s_i$ by $F_i$.
 |  | Let $\hat{\nu}_i := \max_{a' \in \mathcal{A}} \text{ActionValue}(F_i, a')$.
 |  | Let $\hat{p}_i := |F_i|/|F|$.
 | **return** $\sum_{i \in I} \hat{p}_i \left( r(s, a, s'_i) + \gamma \hat{\nu}_i \right)$

---

**Algorithm 3:** ASOP

---

**Data**: The initial state $s_0$, a per-tree budget $b \in \mathbb{N}$ and the forest size $m \in \mathbb{N}$
**Result**: An action to take
**for** $i = 1, \ldots, m$ **do**
 | Let $T_i := \text{SOP}(s_0, b)$.
**return** $\operatorname{argmax}_{a \in \mathcal{A}} \text{ActionValue}(\{T_1, \ldots, T_m\}, a)$

---

## 5 Finite-sample analysis

In this section, we provide a finite-sample analysis of ASOP in terms of the number of planning trees $m$ and per-tree budget $n$. An immediate consequence of this analysis is that ASOP is consistent: the action returned by ASOP converges to the optimal action when both $n$ and $m$ tend to infinity.

Our loss measure is the "simple" regret, corresponding to the expected value of first playing the action $\hat{\pi}(s_0)$ returned by the algorithm at the initial state $s_0$ and acting optimally from then on, compared to acting optimally from the beginning:

$$\mathcal{R}_{n,m}(s_0) = Q^*(s_0, \pi^*(s_0)) - Q^*(s_0, \hat{\pi}(s_0)).$$

First, let us use the "safe" part of SOP to show that each S3-tree is fully explored up to a certain depth $d$ when given a sufficiently large per-tree budget $n$.

**Lemma 1.** *For any $d \in \mathbb{N}$, once a budget of $n \geq 2|\mathcal{A}| \frac{|\mathcal{A}|^{d+1}-1}{|\mathcal{A}|-1}$ has been spent by SOP on an S3-tree, the state-actions of all nodes up and including those at depth $d$ have all been sampled exactly once.*

*Proof.* A complete $|\mathcal{A}|$-ary tree contains $|\mathcal{A}|^l$ nodes in level $l$, so it contains $\sum_{l=0}^{d} |\mathcal{A}|^l = \frac{|\mathcal{A}|^{d+1}-1}{|\mathcal{A}|-1}$ nodes up to and including level $d$. In each of these nodes, $|A|$ actions need to be explored. We complete the proof by noticing that SOP spends at least half of its budget on shallowest leaves. $\square$

Let $v_\omega^\pi$ and $v_{\omega,n}^\pi$ denote the value functions for a policy $\pi$ in the infinite, completely explored S3-tree defined by a random realization $\omega$ and the finite S3-tree constructed by SOP for a budget of $n$ in the same realization $\omega$, respectively. From Lemma 1 we deduce that if the per-tree budget is at least

$$n \geq 2 \frac{|\mathcal{A}|}{|\mathcal{A}|-1} \left[ \epsilon(1-\gamma) \right]^{-\frac{\log|\mathcal{A}|}{\log(1/\gamma)}} . \tag{3}$$

we obtain $|v_\omega^\pi(s_0) - v_{\omega,n}^\pi(s_0)| \leq \left| \sum_{i=d+1}^{\infty} \gamma^i r_i \right| \leq \frac{\gamma^{d+1}}{1-\gamma} \leq \epsilon$ for *any* policy $\pi$.

ASOP aggregates the trees and computes the optimal policy $\hat{\pi}$ of the resulting empirical MDP whose transition probabilities are defined by the frequencies (over the $m$ S3-trees) of transitions from state-action to successor states. Therefore, $\hat{\pi}$ is actually a policy maximizing the function

$$\pi \mapsto \frac{1}{m} \sum_{i=1}^{m} v_{\omega_i,n}^\pi(s_0). \tag{4}$$

If the number $m$ of S3-trees and the per-tree budget $n$ are large, we therefore expect the optimal policy $\hat{\pi}$ of the empirical MDP to be close to the optimal policy $\pi^*$ of the true MDP. This is the result stated in the following theorem.

**Theorem 1.** *For any $\delta \in (0,1)$ and $\epsilon \in (0,1)$, if the number of S3-trees is at least*

$$m \geq \frac{8}{\epsilon^2(1-\gamma)^2} \left( \log|\mathcal{A}| \frac{K}{K-1} \left[ \frac{\epsilon}{4}(1-\gamma) \right]^{-\frac{\log K}{\log(1/\gamma)}} + \log(4/\delta) \right) \tag{5}$$

*and the per-tree budget is at least*

$$n \geq 2 \frac{|\mathcal{A}|}{|\mathcal{A}|-1} \left[ \frac{\epsilon}{4}(1-\gamma) \right]^{-\frac{\log|\mathcal{A}|}{\log(1/\gamma)}} , \tag{6}$$

*then $\mathbb{P}\left( \mathcal{R}_{m,n}(s_0) < \epsilon \right) \geq 1 - \delta$.*

*Proof.* Let $\delta \in (0,1)$, and $\epsilon \in (0,1)$ and fix realizations $\{\omega_1, \ldots, \omega_m\}$ of the stochastic MDP, for some $m$ satisfying (5). Each realization $\omega_i$ corresponds to an infinite, completely explored S3-tree. Let $n$ denote some per-tree budget satisfying (6).

Analogously to (3), we know from Lemma 1 that, given our choice of $n$, SOP constructs trees which are completely explored up to depth $d := \lfloor \frac{\log(\epsilon(1-\gamma)/4)}{\log \gamma} \rfloor$, fulfilling $\frac{\gamma^{d+1}}{1-\gamma} \leq \frac{\epsilon}{4}$.

Consider the following truncated value functions: let $\nu_d^\pi(s_0)$ denote the sum of expected discounted rewards obtained in the original MDP when following policy $\pi$ for $d$ steps and then receiving reward zero from there on, and let $\nu_{\omega_i,d}^\pi(s_0)$ denote the analogous quantity in the MDP corresponding to realization $\omega_i$.

Define, for all policies $\pi$, the quantities $\hat{v}_{m,n}^\pi := \frac{1}{m} \sum_{i=1}^{m} v_{\omega_i,n}^\pi(s_0)$ and $\hat{\nu}_{m,d}^\pi := \frac{1}{m} \sum_{i=1}^{m} \nu_{\omega_i,d}^\pi(s_0)$.

Since the trees are complete up to level $d$ and the rewards are non-negative, we deduce that we have $0 \leq v_{\omega_i,n}^\pi - \nu_{\omega_i,d}^\pi \leq \frac{\epsilon}{4}$ for each $i$ and each policy $\pi$, thus the same will be true for the averages:

$$0 \leq \hat{v}_{m,n}^\pi - \hat{\nu}_{m,d}^\pi \leq \frac{\epsilon}{4} \quad \forall \pi. \tag{7}$$

Notice that $\nu_d^\pi(s_0) = \mathbb{E}_\omega[\nu_{\omega,d}^\pi(s_0)]$. From the Chernoff-Hoeffding inequality, we have that for any fixed policy $\pi$ (since the truncated values lie in $[0, \frac{1}{1-\gamma}]$),

$$\mathbb{P}\left( |\hat{\nu}_{m,d}^\pi - \nu_d^\pi(s_0)| \geq \frac{\epsilon}{4} \right) \leq 2e^{-m\epsilon^2(1-\gamma)^2/8}.$$

Now we need a uniform bound over the set of all possible policies. The number of distinct policies is $|A| \cdot |A|^K \cdot \ldots \cdot |A|^{K^d}$ (at each level $l$, there are at most $K^l$ states that can be reached by following a

policy at previous levels, so there are $|A|^{K^l}$ different choices that policies can make at level $l$). Thus since $m \geq \frac{8}{\epsilon^2(1-\gamma)^2}\left[\frac{K^{d+1}}{K-1}\log|\mathcal{A}| + \log(\frac{4}{\delta})\right]$ we have

$$\mathbb{P}\left(\max_\pi |\hat{\nu}_{m,d}^\pi - \nu_d^\pi(s_0)| \geq \tfrac{\epsilon}{4}\right) \leq \tfrac{\delta}{2}. \tag{8}$$

The action returned by ASOP is $\hat{\pi}(s_0)$, where $\hat{\pi} := \operatorname{argmax}_\pi \hat{v}_{m,n}^\pi$.

Finally, it follows that with probability at least $1 - \delta$:

$$\mathcal{R}_{n,m}(s_0) = Q^*(s_0, \pi^*(s_0)) - Q^*(s_0, \hat{\pi}(s_0)) \leq v^*(s_0) - v^{\hat{\pi}}(s_0)$$

$$= v^*(s_0) - \hat{v}_{m,n}^{\pi^*} + \underbrace{\hat{v}_{m,n}^{\pi^*} - \hat{v}_{m,n}^{\hat{\pi}}}_{\leq 0,\text{ by definition of } \hat{\pi}} + \underbrace{\hat{v}_{m,n}^{\hat{\pi}} - \hat{v}_{m,d}^{\hat{\pi}}}_{\leq \epsilon/4,\text{ by (7)}} + \underbrace{\hat{v}_{m,d}^{\hat{\pi}} - \nu_d^{\hat{\pi}}(s_0)}_{\leq \epsilon/4,\text{ by (8)}} + \underbrace{\nu_d^{\hat{\pi}}(s_0) - v^{\hat{\pi}}(s_0)}_{\leq 0,\text{ by truncation}}$$

$$\leq \underbrace{v^{\pi^*}(s_0) - \nu_d^{\pi^*}(s_0)}_{\leq \epsilon/4,\text{ by truncation}} + \underbrace{\nu_d^{\pi^*}(s_0) - \hat{\nu}_{m,d}^{\pi^*}}_{\leq \epsilon/4,\text{ by (8)}} + \underbrace{\hat{\nu}_{m,d}^{\pi^*} - \hat{v}_{m,n}^{\pi^*}}_{\leq 0,\text{ by (7)}} + \tfrac{\epsilon}{2} \leq \epsilon$$

$\square$

**Remark 1.** *The total budget ($nm$) required to return an $\epsilon$-optimal action with high probability is thus of order $\epsilon^{-2-\frac{\log(K|\mathcal{A}|)}{\log(1/\gamma)}}$. Notice that this rate is poorer (by a $\epsilon^{-2}$ factor) than the rate obtained for uniform planning in [2]; this is a direct consequence of the fact that we are only drawing samples, whereas a full model of the transition probabilities is assumed in [2].*

**Remark 2.** *Since there is a finite number of actions, by denoting $\Delta > 0$ the optimality gap between the best and the second-best optimal action values, we have that the optimal arm is identified (in high probability) (i.e. the simple regret is 0) after a total budget of order $\Delta^{-2-\frac{\log(K|\mathcal{A}|)}{\log(1/\gamma)}}$.*

**Remark 3.** *The optimistic part of the algorithm allows a deep exploration of the MDP. At the same time, it biases the expression maximized by $\hat{\pi}$ in (4) towards near-optimal actions of the deterministic realizations. Under the assumptions of theorem 1, the bias becomes insignificant.*

**Remark 4.** *Notice that we do not use the optimistic properties of the algorithm in the analysis. The analysis only uses the "safe" part of the SOP planning, i.e. the fact that one sample out of two are devoted to expanding the shallowest nodes. An analysis of the benefit of the optimistic part of the algorithm, similar to the analyses carried out in [9, 2] would be much more involved and is deferred to a future work. However the impact of the optimistic part of the algorithm is essential in practice, as shown in the numerical results.*

## 6  Numerical results

In this section, we compare the performance of ASOP to OP-MDP [2], UCT [12], and FSSS [17]. We use the (noisy) inverted pendulum benchmark problem from [2], which consists of swinging up and stabilizing a weight attached to an actuated link that rotates in a vertical plane. Since the available power is too low to push the pendulum up in a single rotation from the initial state, the pendulum has to be swung back and forth to gather energy, prior to being pushed up and stabilized.

The inverted pendulum is described by the state variables $(\alpha, \dot{\alpha}) \in [-\pi, \pi] \times [-15, 15]$ and the differential equation $\ddot{\alpha} = (mgl\sin(\alpha) - b\dot{\alpha} - K(K\dot{\alpha} + u)/R)/J$, where $J = 1.91 \cdot 10^{-4}$ kg $\cdot$ m$^2$, $m = 0.055$ kg, $g = 9.81$ m/s$^2$, $l = 0.042$ m, $b = 3 \cdot 10^{-6}$ Nm $\cdot$ s/rad, $K = 0.0536$ Nm/A, and $R = 9.5\ \Omega$. The state variable $\dot{\alpha}$ is constrained to $[-15, 15]$ by saturation. The discrete time problem is obtained by mapping actions from $\mathcal{A} = \{-3\text{V}, 0\text{V}, 3\text{V}\}$ to segments of a piecewise control signal $u$, each 0.05s in duration, and then numerically integrating the differential equation on the constant segments using RK4. The actions are applied stochastically: with probability 0.6, the intended voltage is applied in the control signal, whereas with probability 0.4, the smaller voltage $0.7a$ is applied. The goal is to stabilize the pendulum in the unstable equilibrium $s^* = (0, 0)$ (pointing up, at rest) when starting from state $(-\pi, 0)$ (pointing down, at rest). This goal is expressed by the penalty function $(s, a, s') \mapsto -5\alpha'^2 - 0.1\dot{\alpha}'^2 - a^2$, where $s' = (\alpha', \dot{\alpha}')$. The reward function $r$ is obtained by scaling and translating the values of the penalty function so that it maps to the interval $[0, 1]$, with $r(s, 0, s^*) = 1$. The discount factor is set to $\gamma = 0.95$.

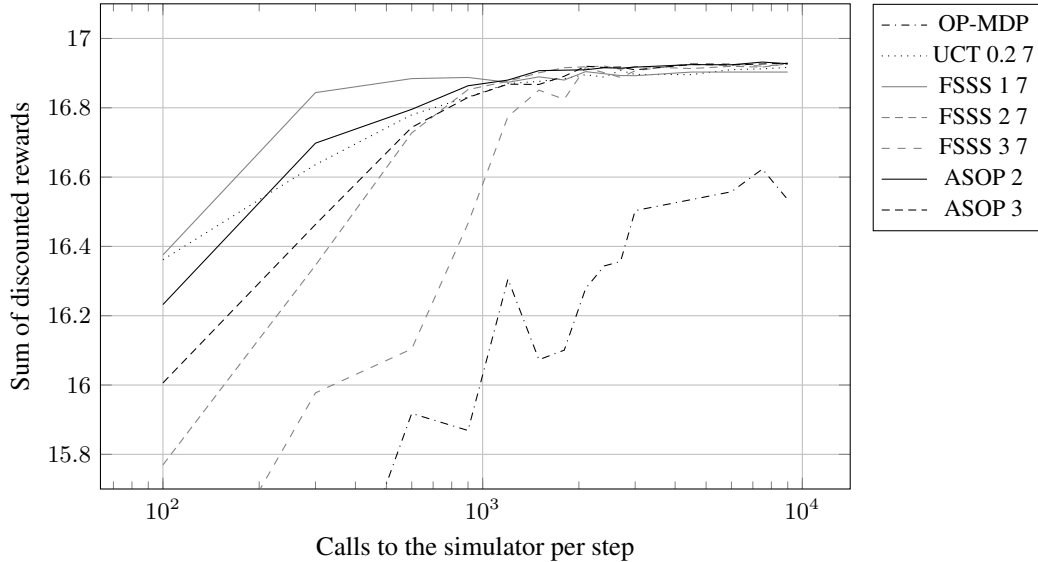

Figure 1: Comparison of ASOP to OP-MDP, UCT, and FSSS on the inverted pendulum benchmark problem, showing the sum of discounted rewards for simulations of 50 time steps.

The algorithms are compared for several budgets. In the cases of ASOP, UCT, and FSSS, the budget is in terms of calls to the simulator. OP-MDP does not use a simulator. Instead, every possible successor state is incorporated into the planning tree, together with its precise probability mass, and each of these states is counted against the budget. As the benchmark problem is stochastic, and internal randomization (for the simulator) is used in all algorithms except OP-MDP, the performance is averaged over 50 repetitions. The algorithm parameters have been selected manually to achieve good performance. For ASOP, we show results for forest sizes of two and three. For UCT, the Chernoff-Hoeffding term multiplier is set to 0.2 (the results are not very sensitive in the value, therefore only one result is shown). For FSSS, we use one to three samples per state-action. For both UCT and FSSS, a rollout depth of seven is used. OP-MDP does not have any parameters. The results are shown in figure 1. We observe that on this problem, ASOP performs much better than OP-MDP for every value of the budget, and also performs well in comparison to the other sampling based methods, UCT and FSSS.

Figure 2 shows the impact of optimistic planning on the performance of our aggregation method. For forest sizes of both one and three, optimistic planning leads to considerably increased performance. This is due to the greater planning depth in the lookahead tree when using optimistic exploration. For the case of a single tree, performance decreases (presumably due to overfitting) on the stochastic problem for increasing budget. The effect disappears when more than one tree is used.

## 7   Conclusion

We introduced ASOP, a novel algorithm for solving online planning problems using a (randomized) simulator for the MDP, under a budget constraint. The algorithm works by constructing a forest of single successor state trees, each corresponding to a random realization of the MDP transitions. Each tree is constructed using a combination of safe and optimistic planning. An empirical MDP is defined, based on the forest, and the first action of the optimal policy of this empirical MDP is returned. In short, our algorithm targets structured problems (where the value function possesses some smoothness property around the optimal policies of the deterministic realizations of the MDP, in a sense defined e.g. in [13]) by using the optimistic principle to focus rapidly on the most promising area(s) of the search space. It can also find a reasonable solution in unstructured problems, since some of the budget is allocated for uniform exploration. ASOP shows good performance on the inverted pendulum benchmark. Finally, our algorithm is also appealing in that the numerically heavy part of constructing the planning trees, in which the simulator is used, can be performed in a distributed way.

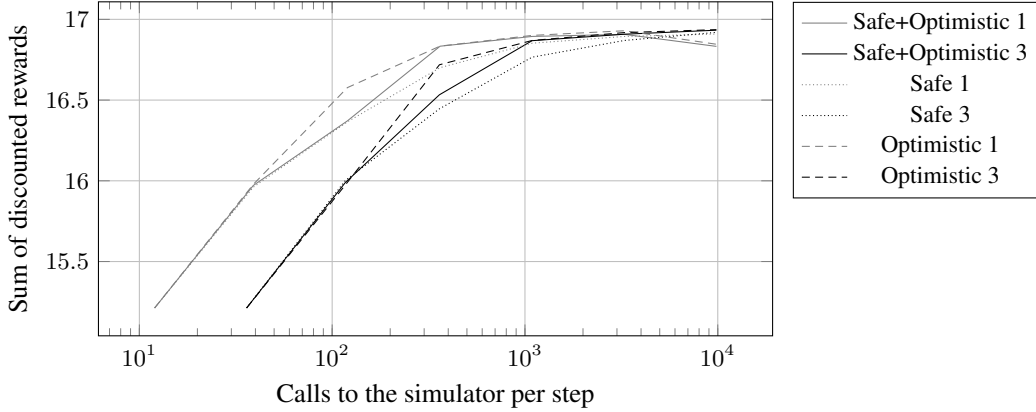

Figure 2: Comparison of different planning strategies (on the same problem as in figure 1). The "Safe" strategy is to use uniform planning in the individual trees, the "Optimistic" strategy is to use OPD. ASOP corresponds to the "Safe+Optimistic" strategy.

## Acknowledgements

We acknowledge the support of the BMBF project ALICE (01IB10003B), the European Community's Seventh Framework Programme FP7/2007-2013 under grant n° 270327 CompLACS and the Belgian PAI DYSCO. Raphaël Fonteneau is a post-doctoral fellow of the F.R.S. - FNRS. We also thank Lucian Busoniu for sharing his implementation of OP-MDP.

## Appendix: Counterexample to consistency when using purely optimistic planning in S3-trees

Consider the MDP in figure 3 with $k$ zero reward transitions in the middle branch, where $\gamma \in (0, 1)$ and $k \in \mathbb{N}$ are chosen such that $\frac{1}{2} > \gamma^k > \frac{1}{3}$ (e.g. $\gamma = 0.95$ and $k = 14$). The trees are constructed iteratively, and every iteration consists of exploring a leaf of maximal b-value, where exploring a leaf means introducing a single successor state per action at the selected leaf. The state-action values are: $Q^*(x, a) = \frac{1}{3}\frac{1}{1-\gamma} + \frac{2}{3}\frac{\gamma^k}{1-\gamma} > \frac{1}{3}\frac{1}{1-\gamma} + \frac{2}{3}\frac{1}{3}\frac{1}{1-\gamma} = \frac{5}{9}\frac{1}{1-\gamma}$ and $Q^*(x, b) = \frac{1}{2}\frac{1}{1-\gamma}$. There are two possible outcomes when sampling the action $a$, which occur with probabilities $\frac{1}{3}$ and $\frac{2}{3}$, respectively:
Outcome I: The upper branch of action $a$ is sampled. In this case, the contribution to the forest is an arbitrarily long reward 1 path for action $a$, and a finite reward $\frac{1}{2}$ path for action $b$.

Outcome II: The lower branch of action $a$ is sampled. Because $\frac{\gamma^k}{1-\gamma} < \frac{1}{2}\frac{1}{1-\gamma}$, the lower branch will be explored only up to $k$ times, as its b-value is then lower than the value (and therefore any b-value) of action $b$. The contribution of this case to the forest is a finite reward 0 path for action $a$ and an arbitrary long (depending on the budget) reward $\frac{1}{2}$ path for action $b$.
For an increasing exploration budget per tree and an increasing number of trees, the approximate action values of action $a$ and $b$ obtained by aggregation converge to $\frac{1}{3}\frac{1}{1-\gamma}$ and $\frac{1}{2}\frac{1}{1-\gamma}$, respectively. Therefore, the decision rule will select action $b$ for a sufficiently large budget, even though $a$ is the optimal action. This leads to simple regret of $\mathcal{R}(x) = Q^*(x, a) - Q^*(x, b) > \frac{1}{18}\frac{1}{1-\gamma}$.

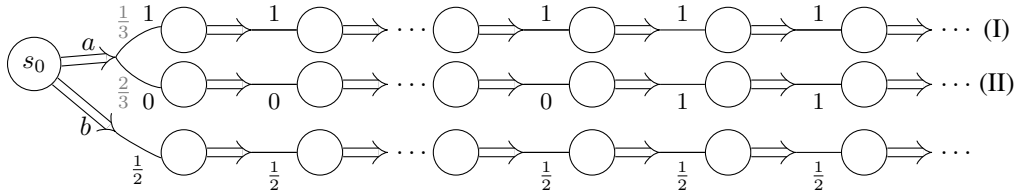

Figure 3: The middle branch (II) of this MDP is never explored deep enough if only the node with the largest b-value is sampled in each iteration. Transition probabilities are given in gray where not equal to one.

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
