[Reviews · NeurIPS 2013]

Submitted by Assigned_Reviewer_4

---Response to Author Response---

Thank you for the clarifications. I hope the authors will consider the following points:

- I'm glad the authors plan to add more comparative results, as they are much needed, but I still see no reason not to compare to UCT. I appreciate the value of comparing to algorithms with similar theoretical properties, but that does not preclude also comparing to the current go-to algorithm in this problem setting. If ASOP does better than UCT in one or more examples, then that is compelling evidence for ASOP's practical applicability as well as its theoretical desirability. If UCT does better then we get a sense of the magnitude of the practical cost of ASOP's desirable guarantees (and can compare that to other algorithms that make similar promises). Neither outcome diminishes ASOP's contribution and either way the paper creates more knowledge for very little extra effort or page space.

- I appreciate the clarification of the relationship between the regret bounds for ASOP and UCT; I hope future versions of this paper will make a similar direct comparison.

- I may be missing something, but I'm not sure I see why it would be difficult to apply Coquelin and Munos' algorithms to your planning problem. "Modified UCT" is essentially just UCT except the exploration bonus is smaller in deeper nodes and it enjoys a singly exponential regret bound with high probability. In any case, the point is just that the paper makes prominent mention of UCT's unfortunate worst case bound, but does not clearly situate ASOP amongst other algorithms in this setting that also improve upon UCT's worst case behavior (even those that are explicitly cited!).

- I now understand better what was meant by "good approximation to the MDP." For me personally, the S3 tree being a good approximation to the MDP suggests that the tree is a close approximation to the MDP model. It sounds like the real point is that the *value function* if the tree is a close approximation to the value function of the MDP. I think it would not be difficult to make this wording more clear/specific in the paper.

- The clarification regarding the best-first expansions helps, and I hope similar clarity will be added to the paper. I personally would find it informative to see an empirical comparison to ASOP with only breadth-first expansions as an illustration of the relative importance of the "optimism" in the algorithm, but if the role of the best-first expansions is clarified in the text, I don't think it would be critical to add this.

---Summary of paper---

The paper presents an online planning algorithm for large MDPs. The main idea is to sample several determinations of the system in the form of roll-out trees where each state/action pair has only one sampled successor. A combination of breadth-first and best-first search is used to explore the deterministic trees, and then they are recombined to create a stochastic model from which a policy can be calculated. The algorithm is proven to be consistent (as the number of trees and number of nodes in each tree both approach infinity, the value at the root can be arbitrarily approximated with high probability). The algorithm is empirically compared to an planning algorithm that requires a full transition model and performs well in comparison.

---Quality---

The analysis provided is, as far as I can tell, correct. I am intrigued by the overall approach, and particularly like the intuition that the S3 trees are "off-policy Monte Carlo" samples. My main concern is with the evaluation of the algorithm. The strengths and weaknesses of this algorithm in comparison to existing approaches to this problem are not very well explored.

The shortcomings of UCT are given as primary motivators for this work, and yet ASOP is not empirically compared to UCT (or to "Modified UCT," the algorithm given in [4] with a singly exponential regret bound with high probability). Also, the regret analysis given here is in terms of the simple regret, but the worst case analysis in [4] is not, making it difficult to compare the two to get a sense of the extent of the improvement (if any) in the bounds. The regret in [4] is related to the number of visits to sub-optimal leaves, and, because it is performing breadth-first search, it seems straightforward to force ASOP to visit exponentially many sub-optimal leaves. So, it is not clear whether this algorithm's theoretical performance guarantees represent an improvement over existing algorithms.

The empirical comparison to OP-MDP seems strange, since it is not even intended for the same problem setting. The fact that OP-MDP requires a full transition model seems to limit the experiment to smallish MDPs, which is counter to the point of ASOP, isn't it? The argument seems to be implied that, because OP-MDP requires access to the transition dynamics, it should have an advantage but, having never heard of it before, I have no idea what kind of performance to expect from OP-MDP nor what to make of the fact that ASOP does better. I can't even tell if either algorithm is doing objectively well, since I don't know what optimal performance would look like in this problem. In all, I did not feel this was an effective or informative empirical demonstration.

The authors should correct me if I am wrong, but it didn't seem like the optimistic part of the algorithm actually played any role in the regret bound. Presumably, then, this part of the algorithm is there to improve empirical performance? If that is the case, it seems like it should be better evaluated. How much do the best-first expansions really help performance? Is it possible for them to hurt performance? Are there another expansion strategy one could consider combining with breadth-first search?

In the conclusions section it is suggested that ASOP should perform best when single-successor trees are good approximations to the original MDP, but I don't see much indication of this in the theory or the experiments. Conceptually, when I think about the extreme case of a deterministic environment, it seems like ASOP would either waste its time creating multiple identical trees or, if it determinism is known a priori, only create one tree but simply perform simultaneous breadth-first and best-first search, which in a lot of cases has done much worse than Monte Carlo tree search algorithms. So I think it would help to have some more justification of this statement.

---Clarity---

For the most part I found the paper to be clear, and the ideas accessible. I appreciate that the high level idea was well-laid out before diving into the details, and that each component of the framework was placed clearly in context in the larger algorithm. I also found the proofs to be mostly easy to follow with a good mix of intuitive and formal argument (though there were some parts that tripped me up, listed below).

- Eq. 1 would be more clear if it was "d = …"

- The proof of Lemma 2 ends in "< \epsilon", but should be "= \epsilon", so the lemma statement should be "<= \epsilon" (adjustments also needed in later statements).

- Shouldn't the exponent in Eq. 3 be ceil[log(\epsilon/4(1 - \gamma))(log \gamma)^{-1}] + 1? I'm not sure where some of those -1s came from.

- Figure 3 is never discussed. What are we meant to learn from it?

There are also several type-os and grammatical mistakes throughout, so the authors should be sure to take some more editing/polishing passes.

---Originality---

The algorithm's conceptual connections to existing work are clearly laid out. I believe the algorithm and its analysis to be novel, and the general approach seems interesting.

---Significance---

The presented algorithm does seem like a new approach to online planning in large domains. It's possible that this approach will turn out to yield significant advances in the state of the art. However, at the end of this paper, I did not feel I had a sense of where this algorithm fits in with existing techniques. It's not that I think every paper needs to be a horse race ending with "Our algorithm is better than that other algorithm." I do think that a new approach, even if it does not immediately compete with existing methods, might still be interesting and valuable to have in the literature. However, there has to be some indication of what that approach brings to the table that is not already there, some indication of what promise it may hold. In this paper, ASOP is presented largely as a novel algorithm in a vacuum, without any comparative analysis (whether theoretical, empirical, or even intuitive). In my opinion that substantially hurts the potential impact of the work.
Summary: I believe the approach is new and the analysis is both clear and correct. However, I did not feel that the strengths and weaknesses of the algorithm were adequately evaluated, particularly in comparison to existing approaches to the same problem. A direct comparative analysis either of theoretical properties or empirical performance would go a long way toward illustrating what this algorithm provides that existing algorithms do not.

Submitted by Assigned_Reviewer_5

The paper describes a novel stochastic simulation technique for planning in MDPs by generating a forest of simple trees and then aggregating the forest into a single tree. This method is claimed to have advantages of depth and breadth searching that other methods do not have. The method is novel, but incremental. The paepr is well written and clear.

The downside of this paper is that only a single algorithm is compared against, and the reasons behind the performance increases are not sufficiently discussed. The paper would be improved if it gave more discussion of *why* this contribution is significant. Why is the method not compared against more generic MCTS planners? What is the motivation for generating all the S3 trees and then aggregating? Why not just generate the aggregate tree (or something similar) directly by a number of states for each action (instead of only one and then aggregating with other sample runs later)? Why is this not compared against?

A few more minor things:
- second paragraph of introduction "made available every for every ..."
- first sentence of section 2 "lead" --> "led"
- figure 1 is not very helpful as it is showing something quite obvious. I would suggest deleting this and adding more discussion of the significance
- last sentence of section 6 : for every values --> for every value
- in bibliography - capitalise Markov
Summary: Overall, the method seems novel and reasonable, but lack of comparisons to other methods, and lack of discussion about what the contributions are, have caused me to lower my score.

Submitted by Assigned_Reviewer_6

Summary:
The authors introduce a new MCTS algorithm based on the aggregation of
trees constructed by the optimism in the face of uncertainty (OFU)
principle. To the best of my knowledge the approach is novel, and I
found it quite interesting. The theoretical analysis seems sound, and
I can believe that the OFU heuristic used by SOP tree construction
process gives a tangible benefit in practice. I also think the work
opens up a number of avenues for future investigation. My only
criticism is that the experimental justification for this approach is
only partially convincing. For example, a comparison of ASOP to FSSS
[18], which enjoys (arguably) comparable theoretical guarantees and
can use a form of branch and bound pruning to speed up the search,
would greatly strengthen the paper in my opinion. Without a more
comprehensive experimental evaluation, I feel this paper is borderline
or at most a weak accept.

Comments:
- Once the aggregate tree is constructed, the final dynamic
programming step to compute the value estimate is quite similar to the
computation performed by sparse sampling. The FSSS algorithm, a sparse
sampling variant which you cite in [18], uses a sound form of branch
and bound pruning to speed up the search. I think this approach could
also be directly applied to speed up your final dynamic programming
step. If this is the case, it is probably worth mentioning and/or
re-running your results to see if it helps.
- Line 138: is the C(x,a) and C(x) needed? Unless I missed it, I don't
think it is used later.
- In Section 5, I think it is worth mentioning that, like sparse
sampling approaches, there is no dependence on |S| in the bounds. Some
further discussion comparing Thm1 to the bounds obtained for Sparse
Sampling [11] would also be interesting.
- In Section 5, it might be helpful to be more specific about what you
mean by a complete tree. It took me a bit of time to work out what was
meant exactly given the assumed property that state-action pairs have
only one successor in the realised trees.

Minor typos:
- Line 41: "... made available every for every" -> is available for every
- Line 42: "... can be referred called" -> is commonly known as
- Line 47: "... (for" -> (short for
- Line 47: "... have allowed to" -> have allowed practitioners to
- Line 169: "... leaves T" -> leaves T_i
Summary: The authors introduce a new MCTS algorithm based on the aggregation of trees constructed via a process that applies the optimism in the face of uncertainty principle. The approach is novel, interesting, and asymptotically consistent. While there is some limited experimental evaluation, it could be more comprehensive.
Author Feedback

Author rebuttal: We thank the 3 reviewers for their very relevant and informative comments.

A major issue that was raised by all 3 reviewers is about numerical experiments: It is true that empirical comparison to UCT as well as other MCTS algorithms is lacking. Our primary goal was to compare ASOP with other planning algorithms that enjoy finite-time performance guarantees, and UCT as well as many other MCTS algorithms may perform well in specific problems but are lacking finite-time theoretical performance guarantees. We chose to compare ASOP to OP-MDP since it applies the optimistic principle in a similar planning setting. We agree that additional experiments are needed and will be added in future version of the work, in particular a comparison with FSSS.

About the difference in the regret definition: ASOP uses the simple regret (like in OP-MDP [2] and OPD [9]) whereas UCT, modified UCT and other algorithms in [4] use a cumulative regret. Although those notions of regret are different in the K-armed bandit setting, an algorithm with R_n cumulative regret can be turned into an algorithm with expected simple-regret of R_n / n (see references [i] and [ii] below).
[i] S. Bubeck, R. Munos, and G. Stoltz. Pure exploration in finitely-armed and continuous-armed bandits. Theoretical Computer Science, 412:1832-1852, 2011
[ii] J.-Y. Audibert, S. Bubeck, and R. Munos. Best arm identification in multi-armed bandits. In Conference on Learning Theory, 2010.

The counter-example mentioned in [4] does not directly fit into the planning scenario considered in our paper. However, a modification of this counter-example in a planning scenario with discounted rewards is possible, and for such problem, the regret of ASOP would be exponential ("only") in the depth D of the planning horizon since ASOP uses half of the numerical budget for expanding the shallowest nodes. This is much better than UCT (which is multiple-times exponential in D) and almost as good as a pure uniform strategy (which would be the best possible strategy when the reward function does not possess any structure). It is not clear how a "modified UCT" could be implemented in this planning problem (while still guaranteeing the B-values to be true upper-bounds).

About the fact that we aggregate at the end and not earlier: One motivation behind ASOP is that the individual planning problems in the S3 trees are deterministic, so the approach "lifts" strategies intended for the deterministic case to the stochastic case. Also, the problem is decoupled; each S3 planning problem can be solved independently of the others.

About the statement that ASOP should perform best in MDPs where S3-trees are "a good approximation to the original MDP": Consider the noisy inverted pendulum used in the numerical evaluation. It has the property that state-action values are somewhat robust against noise: The effect of noise on the control of the pendulum can often be "corrected" by taking an appropriate next action: if the policy (an action sequence) on a particular S3-tree consists of an impulse into a certain direction followed by free movement (a typical situation), and the impulse actually applied is weakened by noise compared to the S3-tree, this can be corrected by consecutively applying another impulse into the same direction instead of the free movement. Therefore, even though the state transitions in the S3-tree may differ from the ones which finally occur, the estimation of the action-value is not too far off; there is a different, "patched" action sequence that has a similar value and the same first action. OP-MDP does not exploit this, and instead considers both cases (the original plan and its correction) separately. As the values are similar, branch-and-bound is not effective in avoiding the duplication of work. For equal budget this leads to deeper, and therefore possibly more precise planning in the case of ASOP.

Some additional answers to reviewer 4:

- It is correct that the "optimistic part of the algorithm" does not play any role in the regret bound. This part of the algorithm is indeed there to improve empirical performance.
- The "optimistic part" helps in our numerical experiments, but it is also true that it may hurt performance. The exploration in each S3 tree by design focusses on an optimal action sequence for this "scenario". Since exploration leads to better lower bounds, those policies which perform well across the considered scenarios are preferred by ASOP. This can be a good thing, but it can also lead to problems, as our counter-example shows. With increasing budget, this bias disappears.
- It is possible to combine the approach with other expansion strategies, maybe incorporating some prior knowledge, but it should be noted that these strategies will also introduce a bias for the reasons described above.
- When using just uniform planning, a swing-up like shown in figure 3 (it gives an example of a computed policy) requires a much higher number of samples; for this problem, a large planning horizon is necessary.
- The assumption of a full transition model in OP-MDP does not imply a restriction to small MDPs, however the number of successor states (the support of the successor state distribution) should not be too large.

Some additional answers to reviewer 6:

- It is correct that the pruning aspect of FSSS could speed up the final dynamic programming step.
- The results for sparse sampling are comparable to those given in Thm1; when each of the S3 trees is fully explored to the same depth H as the planning tree in sparse sampling, and the forest size is C, the action value function has the same near optimality guarantee.